# Study on the Anti-Aging Performance of Different Nano-Modified Natural Ester Insulating Oils Based on Molecular Dynamics

**DOI:** 10.3390/nano13040653

**Published:** 2023-02-07

**Authors:** Tao Zhang, Min Chen, Haohan Zhou, Guangrui Shi, Xiping Fan, Qian Wu

**Affiliations:** 1College of Electrical Engineering and New Energy, China Three Gorges University, Yichang 443002, China; 2Department of Physics and Electronic Information Engineering, Hubei Engineering University, Xiaogan 432000, China

**Keywords:** natural ester insulating oil, aging resistance, molecule simulation, nano-modification, diffusion coefficient, hydrogen bonding

## Abstract

In order to investigate the anti-aging performance of nano-modified natural ester insulating oils, in this paper, two different types of nanoparticles are selected to modify insulating oils. We studied the microscopic mechanism of nano-modified models using molecular simulation techniques. Three models were established: an oil–water model without the addition of nanoparticles and two which contained nano-Fe_3_O_4_ and nano-Al_2_O_3_ particles, where the concentration of water was 1 wt.%. The research found that the diffusion of water molecules in the nano-modified model was slow, and the water molecules generated from transformer insulation aging were adsorbed around the nanoparticles, which inhibited the diffusion of water molecules, reduced the hydrolysis of ester molecules, and effectively enhanced the anti-aging performance of natural ester insulating oil. Compared with two different types of nano-modified models, the interface compatibility between nano-Fe_3_O_4_ and natural ester insulating oil is better, the composite model is stable, the change rate of the diffusion coefficient with temperature is small, there are more hydrogen bonds generated by nano-Fe_3_O_4_ and water molecules, and the anti-aging performance of the nano-Fe_3_O_4_-modified oil model is better.

## 1. Introduction

Transformer insulating oil is used in oil-immersed transformers [1] for arc extinguishing, cooling, heat dissipation and insulation, etc. The performance of the insulating oil determines the transformer’s ability to resist overvoltage insulation, environmental protection, and fire and explosion prevention [2]. To this day, the application of mineral insulating oil in oil-immersed transformers still occupies a dominant position [3]. However, the mineral insulating oil is difficult to degrade, environmentally unfriendly, and non-renewable, indicating the obvious unsustainability of using mineral insulating oil as the main transformer oil. In the past century, a large number of scholars have been actively developing alternative insulating oils while improving the shortcomings of mineral insulating oils [4].

As a new type of liquid dielectric, natural ester is safe, environmentally friendly, heat-resistant, and renewable, and it is now being used in distribution transformers [2]. In order to promote the use of natural ester insulating oil in large transformers of power transmission networks, the following problems of natural ester insulating oil need to be considered: natural ester insulating oil has a high dielectric loss and poor anti-aging ability; the molecular structure of natural ester insulating oil is different from that of mineral insulating oil; and unsaturated bonds exist in the molecules of natural ester insulating oil, which are prone to chemical reactions under the conditions of heat and oxygen, so the anti-oxidation ability is weak. The kinematic viscosity of natural ester insulating oil is large, and the heat dissipation ability is poor compared with that of traditional mineral insulating oil. In order to improve the deficiencies in natural ester insulating oil, scholars have conducted a lot of research and found that the modification of natural ester insulating oil using nanotechnology can effectively improve the performance of insulating oil [5].

It was found that different nanomaterials have different effects on the performance of insulating oils. The authors of [6] prepared four nanofluids of Al_2_O_3_, Fe_3_O_4_, SiO_2_, and SiC for dielectric strength and thermal conductivity measurements; magnetic Fe_3_O_4_ nanomaterials-modified mineral insulating oil can increase the dielectric loss factor of insulating oil [7]; the charge transfer rate and breakdown performance of semiconductor nanoparticle-modified insulating oils were improved [8]; when Fe_3_O_4_ nanoparticles-modified plant insulating oil was used in [9], it was found that the viscosity of the modified oil increased continuously with the increase in the nanoparticle concentration, and the nanoparticles should be modified with surfactants or modified to improve the dispersion stability; experiments verified that nano-Al_2_O_3_-modified insulating oils are better than pure oils in terms of the thermal conductivity and viscosity of nano-transformer oil [10]; the authors of [11] conducted a systematic study on the physicochemical and dielectric properties of Fe_3_O_4_ nano-plant insulating oil; the authors of [12] investigated the structural evolution of cement paste with nano-Fe_3_O_4_ under a magnetic field and found the influence law of nano- Fe_3_O_4_ particles on the structure formation of cement paste; the properties and relaxation times of Fe_3_O_4_, ZnO, Al_2_O_3_, and SiO_2_ nanoparticle materials were summarized in [13]; the authors of [14] compared the effect of Fe_3_O_4_ and Al_2_O_3_ nanoparticles at different concentrations on the dielectric strength of transformer oil at various concentrations; regarding the study of the dielectric breakdown performance of transformer nano-modified insulating oil, the authors of [15,16], respectively, prepared Fe_2_O_3_ magnetic nanofluid and T-oil-based magnetic fluid containing Mn-Zn ferrite nanoparticles for experiments, and they found that the presence of nanoparticles could significantly improve the dielectric breakdown performance of transformer insulating oil.

By comparing the modification effects of different types of nanoparticles, it is easy to find that the comprehensive modification effect of magnetic nano-insulating oil prepared using Fe_3_O_4_ nanoparticles is outstanding. Due to their unique magnetic and electrical properties, Fe_3_O_4_ nanoparticles have considerable research value in the study of transformer insulating oil modification [17]. The magnetic Fe_3_O_4_-nano-modified oil exhibits a better performance than crude oil in terms of dielectric properties, breakdown field strength, and thermal conductivity. In order to accelerate the application of nano-modified insulating oil in large transformers, the aging resistance of nano-insulating oil needs to be further verified [18].

In order to investigate the effect of magnetic Fe_3_O_4_ nanoparticles on the performance of aged natural ester insulating oil, the insulating oil is first aged. Water molecules can degrade the insulation performance within the transformer, and adding a certain percentage of water molecules to the crude oil can accelerate the aging of the insulating oil. In this paper, the molecular dynamics simulation technique is used to calculate the natural ester insulating oil before and after nano-modification, where the nano-materials are selected with reference to [14], in which the non-magnetic material Al_2_O_3_ and the magnetic material Fe_3_O_4_ are chosen for comparative analysis to compare the performance of the two nano-modified oils and to explore the mechanism of nano-particle modification.

## 2. Interaction of Natural Ester Insulating Oil with Metal Oxide Surfaces

Metal nanomaterials have the characteristics of a large specific surface area, many surface-active centers, and increased surface-active sites. After doping the metal nanoparticles into the natural ester insulating oil, the nanoparticles and the insulating oil molecules will form a two-phase interface, and the interfacial interaction energy has a great influence on the stability of the nano-modified oil [19]. If the interaction energy between the surface of insulating oil molecules and the surface of nanoparticles is positive, it means that the two substances are incompatible; if the interaction energy is negative, it indicates that the two substances are attracted to each other and that the interface is compatible. The larger the absolute value of the negative value of the interfacial interaction energy, the better the interfacial compatibility of the two substances and the better the stability of the nano-modified oil.

The modified nanomaterials Fe_3_O_4_ and Al_2_O_3_ selected in this paper are both metal nanomaterials. In order to initially determine whether the metal nanoparticles are compatible with the natural ester insulating oil, and to compare the stability of the two nano-modified oils, we established an interface composite model between metal oxide and natural ester insulating oil and calculated the interaction energy between insulating oil molecules and the metal oxide interface.

### 2.1. Natural Ester Oil-Based Model Building

Natural ester insulating oil for domestic and foreign commercial transformers is generally made from vegetable oils such as soybean oil, rapeseed oil, palm oil, and other vegetable oils as base materials, and refined natural ester insulating oil is prepared through a series of optimized processes such as alkali refining, deodorization, and deacidification. The main component of natural ester insulating oil is TriacylGlycerol, which is an organic compound generated by the esterification of three hydroxyl groups of glycerol and three fatty acid molecules, where the type of fatty acid in vegetable crude oil determines the main performance of the natural ester insulating oil. The type of fatty acid can be judged according to the degree of the saturation of hydrocarbon chains. Saturated fatty acids do not contain carbon–carbon double bonds in their molecular formula and are chemically stable, but they have a relatively high freezing point. Unsaturated fatty acids contain one or more carbon–carbon double bonds in their molecular formula, and these fatty acids have relatively low freezing points and a low viscosity, but they are chemically unstable and have poor oxidative stability. The physicochemical and electrical properties of the different vegetable oil-based natural ester insulating oils prepared were compared with those of mineral insulating oils, and it was found that the best overall performance was exhibited by the natural ester insulating oils prepared with a higher content of monounsaturated fatty acids in an vegetable oil base [20]. Table 1 shows the content of fatty acid components in various vegetable oil bases [21].

In Table 1, it can be seen that the saturated fatty acid content of the rapeseed oil base is about 7.9%, mainly composed of stearic acid, the monounsaturated fatty acid content is about 55.9%, mainly composed of oleic acid, the diunsaturated fatty acid content is about 22.1%, mainly composed of linoleic acid, and the triunsaturated fatty acid content is about 11.1% of the main component, which is α-linoleic acid. In comparing the fatty acid content with the actual production cost, rapeseed oil was selected as the raw oil for the preparation of natural ester insulating oil. Canola oil-based natural ester insulating oil is a triglyceride produced by the esterification and dehydration of fatty acid and glycerol molecules, so different types of triglyceride molecular models can be constructed as glyceryl tristearate, glyceryl trioleic acid, glyceryl trinoleic acid, and glyceryl trinolenic acid [22]. The mass percentage of triglycerides can be inferred from the fatty acid component content of rapeseed oil in Table 1, and 100 g of rapeseed oil-based natural ester insulating oil was taken. The ratio of different types of the natural ester oil-based triglycerides model can be obtained as 1:7:3:2 by molar calculation, and the natural ester insulating oil model was constructed using the method proposed by Theodorou [23] for constructing amorphous polymers, as shown in Figure 1.

### 2.2. Fe_3_O_4_ and Al_2_O_3_ Surface Model Building

When building the molecular model, the force field and charge should be set correctly first, and the COMPASS II force field suitable for metal oxides is used. It should be emphasized that Fe_3_O_4_ can be approximated as a compound composed of FeO and Fe_2_O_3_, which means that the force field types of Fe atoms in the Fe_3_O_4_ cell model are divided into two types: one is applicable to FeO, and all +2 valence Fe atoms are selected and set to fe2o; the other is applicable to Fe_2_O_3_, and all +3 valence Fe atoms are selected and set to fe3o, as shown in Figure 2.

In order to reduce the calculation error caused by the different size of the model surface, the lengths of the model dimensions U and V need to be viewed after cutting the crystal surface, where the parameters U and V have equal values. In cutting the (1 0 0) crystal surface of the unit cell Fe_3_O_4_ and the (0 0 −1) crystal surface of the unit cell Al_2_O_3_, the thickness of the surface layer is 13Å, and the lengths of the model U and V are shown in Figure 3. Since the surface area of the original unit cell cut surface is small, the calculated surface area is increased according to the lengths of the surface model dimensions U and V. The supercell surface models of Fe_3_O_4_ (4 × 4) and Al_2_O_3_ (5 × 5) are respectively established for the construction of composite interface models.

### 2.3. Construction of Layer Structure Model Building

The Build layers module in Materials Studio was used to construct the surface composite model of natural ester insulating oil and metal oxide. In layer 1 the relaxed metal oxide (4 × 4) surface was selected; in layer 2 the natural ester insulating oil amorphous monocell was selected, the vacuum layer of layer 2 was added to 30Å in the Layer Details module, and this composite model was named NG-Fe_3_O_4_. The abovementioned operation was repeated to construct another composite model named NG-Al_2_O_3_ by selecting the relaxed Al_2_O_3_ (5 × 5) surface in layer 1. The model structure is shown in Figure 4.

### 2.4. Simulation Details

In this paper, when performing molecular dynamics calculations on the constructed layer structure model, model optimization is to be performed first, using the Fix Cartesian position function to fix all atoms of the metal oxide and using the COMPASSII force field with the Max.iteration set to 5000 when optimizing the structure. Keeping the metal oxide atoms still fixed, the optimized layer structure model was calculated by molecular dynamics, using NVT regular system synthesis with an integration step of 1fs. The transformer is mainly affected by the temperature and electric field during operation [24]. In order to investigate the influence of the temperature and electric field on the stability of the composite model, and to compare the stability differences between the two composite models of NG-Fe_3_O_4_ and NG-Al_2_O_3_, it is necessary to set reasonable temperatures and electric fields for molecular dynamics calculations. During the actual operation of the transformer, its internal oil temperature is approximately between 70 °C and 120 °C [25], and the hot spot temperature can be from 120 °C to 140 °C. Therefore, the temperature range is set to 70 °C to 150 °C, which is converted to a thermodynamic temperature range of 343 K to 423 K, with an interval of 20 K. The molecular dynamics calculations are performed for the two sets of composite models at five temperatures. Since the simulation environment did not consider the possible breakdown of the material in actual operation, and to shorten the simulation time, the simulated electric field strength was set to an electrostatic field of 10^10^ V/m, with the electric field direction along the positive direction of the Z axis [26]. The Forcite module could not perform the electric field calculation directly, and it needed to be run using a script edited in the perl language. The stability of the interfacial composite model of metal oxide and natural ester insulating oil molecules is related to the interaction energy between the two substances forming the interface, and the interfacial interaction energy is calculated, as shown in Equation (1).

This is example 1 of an equation:(1)$$E_{int}=E_{total}-(E_{substrate}+E_{triacylglycerol})$$

$E_{int}$ is the interaction energy between metal oxides and insulating oil molecules, $E_{total}$ is the total energy of the interface model, $E_{substrate}$ is the potential energy values for the metal oxides surface model, $E_{triacylglycerol}$ is the potential energy values for the insulating oil molecules model [27].

### 2.5. Results and Discussion

We subjected the composite layer structure models NG-Fe_3_O_4_ and NG-Al_2_O_3_ to molecular dynamics calculations and analyzed the results. Table 2 gives the calculation results of the interaction energy between the two model molecules at different temperatures without the applied electric field, and all the values of interaction energy are negative, indicating that the two metal oxides selected in this paper are compatible with the insulating oil molecules. In the table, $E_{int}$ is the total interaction energy of the two composite layer structure models, $E_{vdw}$ is the van der Waals interaction energy, and $E_{elec}$ is the electrostatic interaction. From the results, it can be seen that 95% of the interaction energy between the metal oxide and insulating oil molecules comes from the intermolecular van der Waals force and electrostatic force, and the electrostatic interaction energy is dominant.

From Table 2, it is easy to find that the values of $E_{int}$ in the two models do not change much with the change in temperature, indicating that the change in temperature has little effect on the compatibility stability of the composite model. At the same temperature, the values of the model NG-Fe_3_O_4_ interaction energy are much larger than those of the model NG-Al_2_O_3_. The calculated results of the interaction energy between the two model molecules at different temperatures with the applied electric field strength of 1010 V/m are given in Table 3. Comparing the data in Table 2 and Table 3, it can be seen that both model intermolecular interaction energies changed after the application of the electric field, with the electric field having a greater effect on the electrostatic interaction energy $E_{elec}$. The interaction energy between Fe_3_O_4_ and insulating oil molecules under the same temperature and electric field is still greater than that between Al_2_O_3_ and insulating oil molecules. From the change in data, the model NG-Al_2_O_3_ interaction energy increased by 30%, indicating that the composite model is more influenced by the electric field; the model NG-Fe_3_O_4_ intermolecular interaction energy is reduced by less than 10% by the electric field, indicating that the composite model is less influenced by the electric field. In summary, the interaction between Fe_3_O_4_ and insulating oil molecules is stronger, and the composite model is more stable.

## 3. Anti-Aging Analysis of Nano-Modified Natural Ester Insulating Oil

### 3.1. Model Building

In order to compare and analyze the aging resistance of different nano-modified natural ester insulating oils, the natural ester insulating oils are to be aged before and after modification. Water molecules are one of the products of insulation aging in transformers, and moisture has an important influence on the internal insulation property of transformers. We added the two nanoparticles mentioned in the previous section to the aging insulating oil and used molecular dynamics simulation to investigate the effect of nanoparticles on water molecules in the insulating oil and to compare the modification effect of the two nanomaterials on the aging insulating oil. Three models were built: the oil–water model NG-1 without added nanoparticles and two sets of nano-modified models (NG-2 and NG-3) with nano-Al_2_O_3_ and nano-Fe_3_O_4_ particles, respectively, added to the oil–water model, where the water content is 1 wt% of the natural ester insulating oil content. The model construction process is shown in Figure 5, where the particle radius of the Al_2_O_3_ and Fe_3_O_4_ nanoparticle cluster model is 5Å.

### 3.2. Simulation Details

First, the constructed three groups of models, NG-1, NG-2, and NG-3, were optimized for structure, and when the model structure reached equilibrium, the temperature was set to 343 K~423 K and annealed for five cycles under the NVE system synthesis to obtain the steady state model with the lowest energy. The kinetic simulations were carried out after the optimization and annealing of the three models; first, the NVT regular system was used, the simulation time was set to 200 ps, and the temperature was set to 343 K. Later, the isothermal and isobaric system was used to calculate the molecular dynamics under the same parameters to obtain the kinetic trajectory at this temperature, and then the temperature was ramped up to 423 K in order to perform the same calculations. The abovementioned calculations were performed using the COMPASS II force field, the Nosé-Hoover method for temperature control, the Velocity Verlet method for the integration algorithm, the Atom-based method for van der Waals interaction, and the Ewald method for electrostatic interaction. It is important to emphasize that, although the charge and force fields have been set for each of the three models before the calculation, the Fe_3_O_4_ nanoparticles still contain invalid floating points in the structure and need to be fixed for the kinetic calculation under the NVT regular system synthesis.

### 3.3. Results and Analysis

#### 3.3.1. Diffusion Coefficient of Water Molecules

The water molecules in the insulating oil diffusion will accelerate insulation aging, increase the dielectric loss, and reduce the insulation performance, resulting in shortening the service life of the transformer. The diffusion motion of water molecules in insulating oil is influenced by temperature [28], and its motion can be expressed by the mean square displacement (MSD), which describes the average of the distance traveled by the particle at moment t [29,30]; the relationship is (2).

This is example 2 of an equation:(2)$$MSD=\langle {\left|r_{i}(t)-r_{i}(0)\right|}^{2}\rangle$$

The analytical calculation of the MSD curves of the water molecules for the three groups of models (NG-1, NG-2, and NG-3) under the action of different temperatures is shown in Figure 6.

The simulation time is 200 ps for all three groups of models, and we only show the MSD curves for water molecules from 0 to 150 ps. It can be seen from the figures that the range of motion of water molecules in the three groups of models is different at different temperatures. As the temperature gradually increases, there is a large difference in the range of motion of water molecules in the unmodified model and the two sets of nano-modified models. Figure 6a shows the MSD curves of the unmodified model NG-1. From the figure, it can be seen that, when the temperature is 343 K, the mean square displacement size of water molecules varies in the range of 0–50 Å^2^; when the temperature is 363 K~403 K, the mean square displacement of water molecules varies in the range of 0–200 Å^2^; when the temperature is 423 K, the mean square displacement of water molecules varies in the range of 0–300 Å^2^ varies. Figure 6b,c show the MSD curves of the Al_2_O_3_ and Fe_3_O_4_ nano-modified models, respectively, compared with the unmodified model curves; the variation in the mean square displacement of water molecules in both nano-modified models is smaller than that of the NG-1 model at the same temperature, and the maximum range of motion does not exceed 100 Å^2^. The nanoparticle Al_2_O_3_-modified oil showed that the mean square displacement of water molecules varied in the range of 0–75 Å^2^; the mean square displacement of water molecules in the NG-3 nanoparticle Fe_3_O_4_-modified oil varied in the range of 0–50 Å^2^. At different temperatures from 343 K to 423 K, the mean square displacement of water molecules in the model NG-3 varied less than that in the model NG-2, indicating that the thermal stability performance of the Fe_3_O_4_ nanoparticle-modified oil was better than that of the Al_2_O_3_ nanoparticle-modified oil. This indicates that the diffusion of water molecules is relatively slow at low temperatures, and the diffusion range of water molecules gradually increases as the temperature increases. The addition of nanoparticles can effectively inhibit the effect of temperature on the diffusion of water molecules, thus reducing the risk of thermal ageing and enhancing the thermal stability of natural ester insulating oils.

There is a correspondence between the magnitude of MSD (mean square displacement) and the diffusion coefficient of atoms, and the diffusion coefficient of particles (*D*) is calculated according to Einstein’s diffusion law with the relation (3).

This is example 3 of an equation:(3)$$D=\frac{1}{6N}\underset{t\to \infty }{\lim}\frac{d}{dt}\sum_{t=1}^{N}\langle {\left|r_{i}(t)-r_{i}(0)\right|}^{2}\rangle$$
where *D* is the diffusion coefficient of the particle, and *N* is the number of diffusing particles in the system. When the time is longer, the slope of the mean square displacement curve is six times the diffusion coefficient, from which the diffusion coefficient of water molecules in the three groups of models can be obtained, as shown in Table 4.

Comparing the values in the table, we can see that the diffusion coefficients of water molecules in the three groups of models gradually increase with the increase in temperature. At the same temperature, the diffusion coefficients of water molecules in the two groups of oil models containing nanoparticles are smaller than those in the NG-1 crude oil water model, which indicates that the movement of water molecules is slowed down by the influence of nanoparticles, thus proving that nanoparticles have a certain binding ability to water molecules, but the binding ability of nanoparticles to water molecules gradually decreases with the increase in temperature. Comparing the values of the nano-modified oil models NG-2 and NG-3, it can be seen that the model NG-3 has a smaller diffusion coefficient of water molecules under the effect of different temperatures, which indicates that Fe_3_O_4_ nanoparticles have a stronger binding effect on water molecules.

#### 3.3.2. Free Volume

The free volume is one of the main factors affecting the diffusion of particles in the medium, and the diffusion phenomenon can be described by the free volume theory [31,32]; the diffusion coefficient D is related to the free volume $V_{FV}$ as (4).

This is example 4 of an equation:(4)$$D=ke^{-\gamma V^{*}/V_{FV}}$$

k is a factor relating the particle kinetic rate, the geometry factor, and the molecular size; γ is the overlapping free volume correction factor; $V^{*}$ is the critical free volume for diffusion; $V_{FV}$ is the temperature- and pressure-dependent free volume, further indicating that the diffusion coefficient increases with the increasing temperature. The three sets of models were subjected to molecular dynamics calculations at temperatures from 343 K to 423 K to obtain the free volume of the insulating oil under the action of different temperatures, and Figure 7 shows the free volume of Al_2_O_3_ nano-modified oil at a temperature of 343 K.

The volume of the insulating oil liquid dielectric can be divided into two parts: occupied volume and free volume. In Figure 7, the blue part is the occupied volume, which is the space occupied by the molecules themselves, and the gray part is the free volume, which is the space between molecules. The diffusion of particles and the size of the free volume are positively correlated, the free volume fraction (FFV) is equal to the ratio of the free volume of the medium to the total volume, and the calculation formula is shown in Equation (5), from which the free volume fraction of the three groups of models at different temperatures can be obtained, as shown in Figure 8.

This is example 5 of an equation:(5)$$FFV=\frac{V_{FV}}{V_{OF}+V_{FV}}$$

$V_{OV}$ is the occupied volume, and $V_{FV}$ is the free volume.

It can be seen from the graph in Figure 8 that the free volume fractions of all three groups of models show an increasing trend at different temperatures. At the same temperature, the free volume fraction of the model NG-1 was much larger than that of the two nano-oil models, and the growth rate of the free volume fraction of the model NG-1 was the fastest as the temperature increased. The growth rate of the model NG-1 free volume fraction was calculated to be about 9.5%, that of the model NG-2 free volume fraction was about 5.6%, and that of the model NG-3 free volume fraction was about 4.4%. It is obvious that the nanoparticles added into the insulating oil reduce the growth rate of the free volume fraction in the oil mold and decrease the effect of the temperature change on the diffusion behavior of water molecules, and the growth rate of the free volume fraction of the model NG-3 modified by nano-Fe_3_O_4_ is the smallest, which indicates that the model NG-3 nano-Fe_3_O_4_-modified natural ester insulating oil has better thermal stability. The maximum value of the free volume fraction was observed when the operation temperature increased to 423 K, which explained the trend of the MSD curve of the water molecules in the model at this temperature.

#### 3.3.3. Hydrogen Bonding

By calculating and analyzing the water molecule diffusion coefficient and free volume fraction of the three groups of models, it is found that the two groups of nano-modified models have a better thermal stability performance and a smaller growth rate of the free volume fraction with temperature, and the diffusion coefficient of water molecules in the models is smaller than that in the model of insulating oil without nanoparticles added. In order to investigate the mechanism of the influence of nanomaterials on the diffusion movement of water molecules and to determine the molecular dynamics of water molecules, nanoparticles, and natural ester insulating oil at different temperatures, the binding free energy of the three groups of model molecules should be calculated. The binding free energies between small molecules are mainly non-bonding interactions, including van der Waals interaction, electrostatic interaction, and hydrogen bonding. Among them, hydrogen bonding is a stronger non-bonding interaction between hydrogen atoms and atoms with a larger electronegativity, and hydrogen bonds exist both within and between molecules. The number of hydrogen bonds and the average bond lengths after counting the molecular dynamics simulations of the three groups of models at 363 K are shown in Table 5.

From the results in Table 5, it can be seen that the addition of nanoparticles increased the number of hydrogen bonds, and the average bond lengths were smaller. Compared with the two groups of nano-modified models, the average bond length of hydrogen bonds in the NG-3 model Fe_3_O_4_ nano-modified oil is smaller than the average bond length of hydrogen bonds in the NG-2 model Al_2_O_3_ nano-modified oil, indicating that the model NG-3 is more stable. Since hydrogen bonds have saturation, the number of hydrogen bonds in the two groups of nano-modified insulating oil models is the same, but the average bond lengths of hydrogen bonds are different. In order to further investigate the effects of the two nanoparticles on water molecules, all the hydrogen bonds in the models are shown in Figure 9.

A hydrogen bond is the covalent bonding of an H atom to an X atom with a large electronegativity. If another atom Y with a large electronegativity and a small radius is present, a special intramolecular or intermolecular interaction is formed between X and Y using the H atom as a medium to produce X-H…Y, where X and Y represent non-metallic atoms with a large electronegativity and a small atomic radius, such as O atoms, F atoms, and N atoms. As can be seen in Figure 9, hydrogen bonds are generated in the three models mainly between water molecules and water molecules, between water molecules and triacylglycerol, and between water molecules and nanoparticles, generating O-H…O. Figure 9a shows the types of hydrogen bonds in the unmodified oil–water model, with 13 hydrogen bonds generated in the last frame of the simulation, when nanoparticles are not added to the model, mainly generated by the H atom in the water molecule and the O atom in the triacylglycerol. The hydrogen bonds generated by water molecules and triacylglycerol form a special intermolecular interaction, which accelerates the hydrolysis reaction of ester molecules in natural ester insulating oil, thus reducing the insulating property of natural ester insulating oil.

During the playback of the simulated track, it was found that the water molecules in the model NG-1 had the largest range of motion, while the water molecules in the two nano-modified models NG-2 and NG-3 almost surrounded the nanoparticles. Figure 9b,c, respectively, show the types of hydrogen bonds in the nano-Al_2_O_3_ and nano-Fe_3_O_4_ modification models, and it can be seen that hydrogen bonds are generated between the water molecules and the nanoparticles. Comparing the types of hydrogen bonding in the two modified models, the model NG-2 shows that the H atoms in the water molecules form hydrogen bonds with the O atoms in the nanoparticles as well as with the O atoms in the natural ester insulating oil. On the other hand, the model NG-3 in the last frame type of hydrogen bonding is seen almost exclusively between the H atoms in the water molecules and the O atoms in the nanoparticles. Therefore, both the type of the hydrogen bond and the bond length size of the hydrogen bond differ, with the average bond length of the hydrogen bond being shorter in the nano-Fe_3_O_4_ model. There is an obvious effect of the O atoms in the nanoparticles forming hydrogen bonds with the H atoms in the water molecules in the NG-3 model, and this indicates that the Fe_3_O_4_ nanoparticles have a greater influence on the water molecules and is the reason why the Fe_3_O_4_ nanoparticles are better able to inhibit the diffusion of water molecules. Figure 10 shows the distribution of water molecules in the nano-Fe_3_O_4_-modified model at the 101st ps of the simulation.

## 4. Conclusions

In this paper, different types of nanomaterials were selected. The stability and anti-aging properties of different nano-modified oils were studied micro-mechanically using molecular simulation techniques, and several conclusions were obtained, as follows:(1)Fe_3_O_4_ and Al_2_O_3_ metal oxide molecules form two different sets of interfacial models with insulating oil molecules, and the interfacial interaction energies of the two composite models are calculated under the action of the temperature and the electric field, respectively. The results show that the interaction energy of the model NG-Fe_3_O_4_ interface is larger than that of the model NG-Al_2_O_3_ under the action of different temperature fields with or without an electric field; this indicates that the Fe_3_O_4_ surface model and the insulating oil molecules, with obvious attraction, have better interfacial compatibility, and the composite model is more stable.(2)The diffusion coefficients of water molecules and the free volumes of the models in the unmodified oil–water model and the nano-modified oil–water model were calculated.

From the trend of the water molecule MSD curves in several groups of models, it can be seen that the range of water molecule diffusion movement increases with the increase in temperature, which is because the free volume in the model increases to provide movement space for water molecule diffusion. However, the free volume fractions of the nano-models of NG-2 and NG-3 are smaller at the same temperature, and the growth rate of the free volume fraction with the increase in temperature is smaller than that of the model NG-1, which indicates that the nanoparticles slow down the effect of temperature on the diffusive motion of water molecules in insulating oil. The diffusion range of water molecules and the change rate of the free volume fraction in the NG-3 model are smaller than those in the NG-2 model, which indicates that the binding effect of Fe_3_O_4_ on water molecules is stronger and that the modified model has better thermal stability.

(3)By calculating the number of hydrogen bonds and the average bond length of the two modified models, it was found that the number of hydrogen bonds between water molecules and other molecules was the same in the two modified models, but the types of hydrogen bonds and the average bond length of hydrogen bonds were different. In the nano-Fe_3_O_4_ model, the O atoms in the nanoparticles form hydrogen bonds with the H atoms in the water molecules, and the bond lengths are shorter, indicating that the Fe_3_O_4_ nanoparticles have more influence on the water molecules.

## Figures and Tables

**Figure 1 nanomaterials-13-00653-f001:**
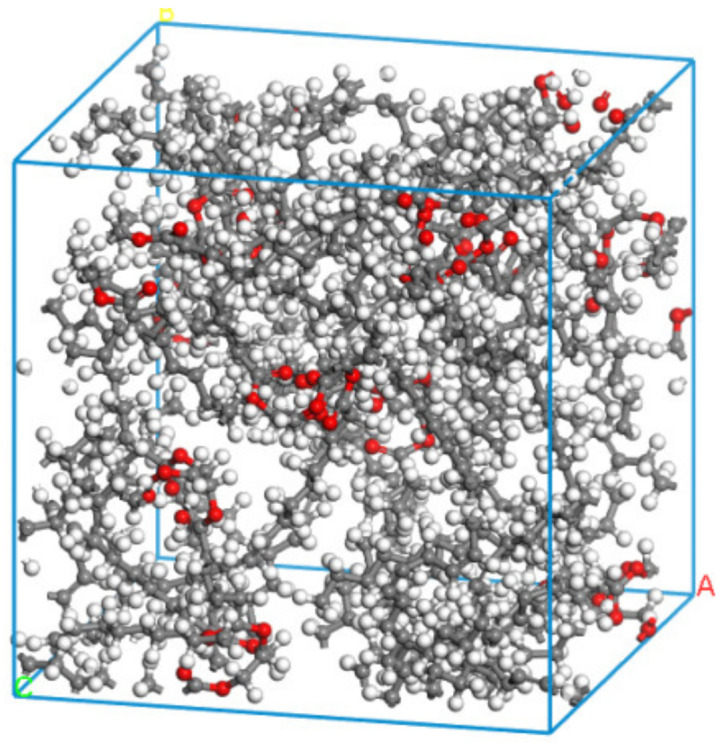
Model of natural ester insulating oil.

**Figure 2 nanomaterials-13-00653-f002:**
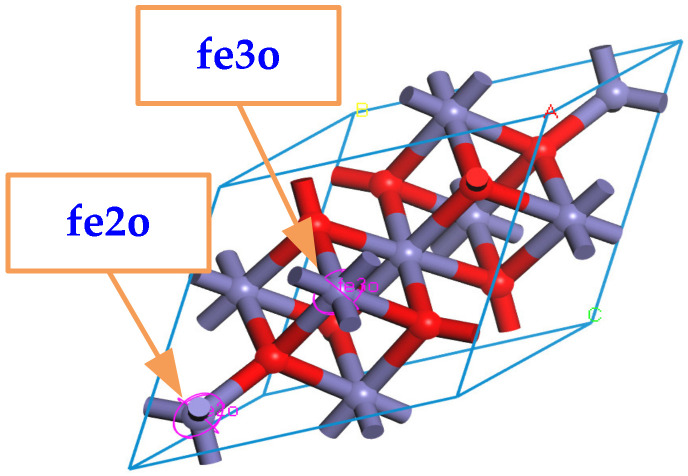
Multiple forcefield types for Fe atoms in Fe_3_O_4_.

**Figure 3 nanomaterials-13-00653-f003:**
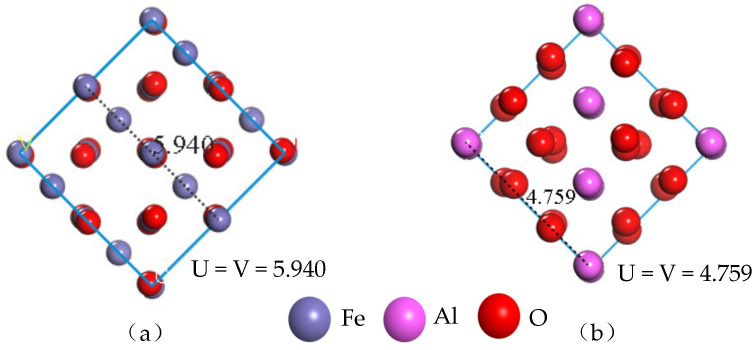
Lengths of the original unit cell cut surface U and V: (**a**) Fe_3_O_4_ surface model, (**b**) Al_2_O_3_ surface model.

**Figure 4 nanomaterials-13-00653-f004:**
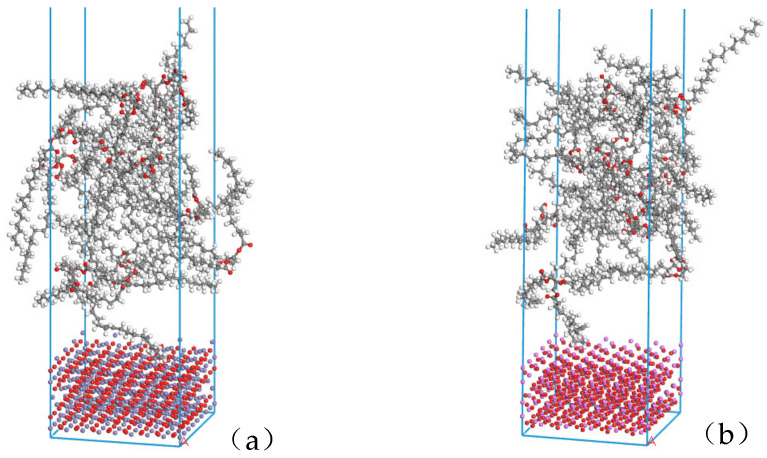
Composite structure models: (**a**) Fe_3_O_4_ with the insulating oil layer structure model NG-Fe_3_O_4_, (**b**) Al_2_O_3_ with the insulating oil layer structure model NG-Al_2_O_3_.

**Figure 5 nanomaterials-13-00653-f005:**
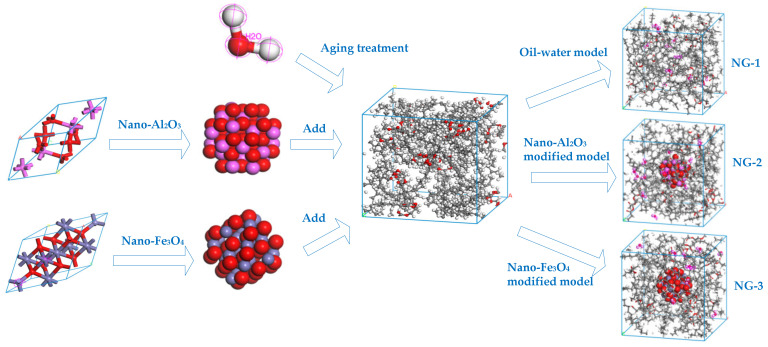
Construction process of the computational models before and after modification.

**Figure 6 nanomaterials-13-00653-f006:**
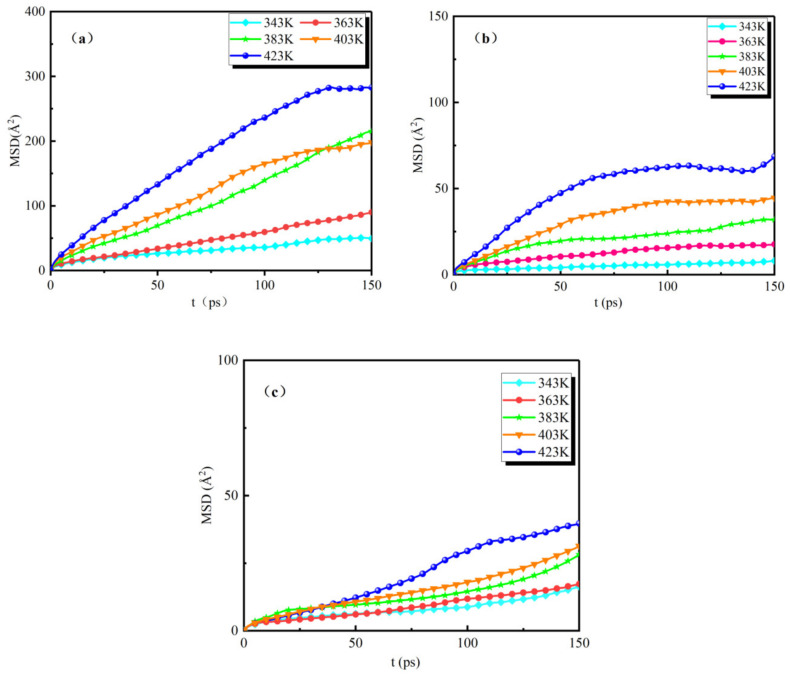
MSD curves of water molecules in three groups of models at different temperatures: (**a**) MSD curves of the oil–water model, (**b**) MSD curves of the Al_2_O_3_ nano-modified model, (**c**) MSD curves of the Fe_3_O_4_ nano-modified model.

**Figure 7 nanomaterials-13-00653-f007:**
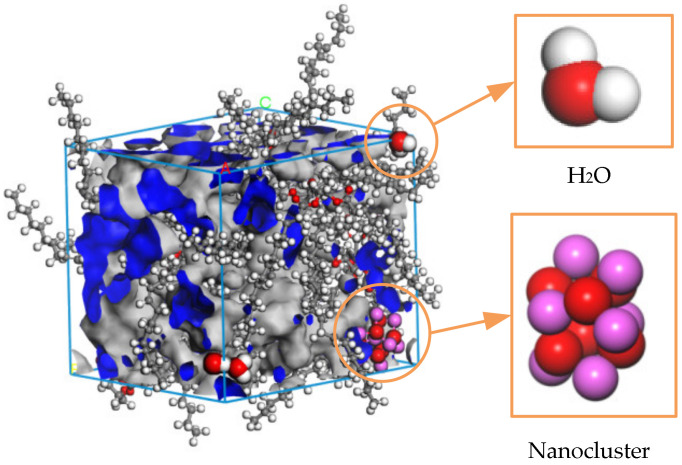
Free volume of the Al_2_O_3_ nano-modified model at 343 K.

**Figure 8 nanomaterials-13-00653-f008:**
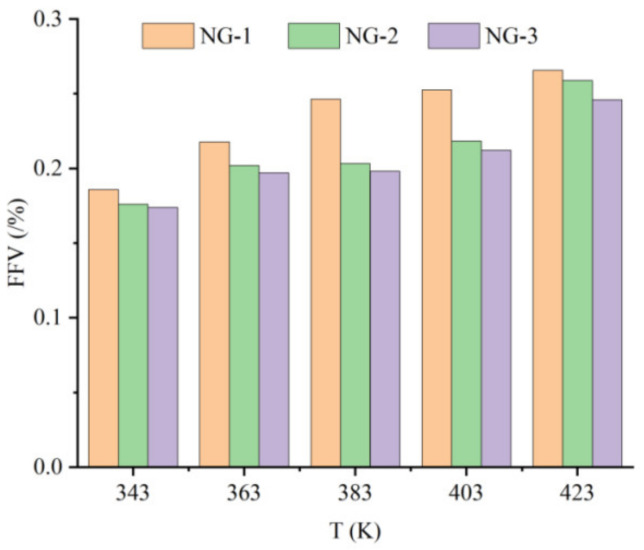
Free volume fraction of three models at different temperatures.

**Figure 9 nanomaterials-13-00653-f009:**
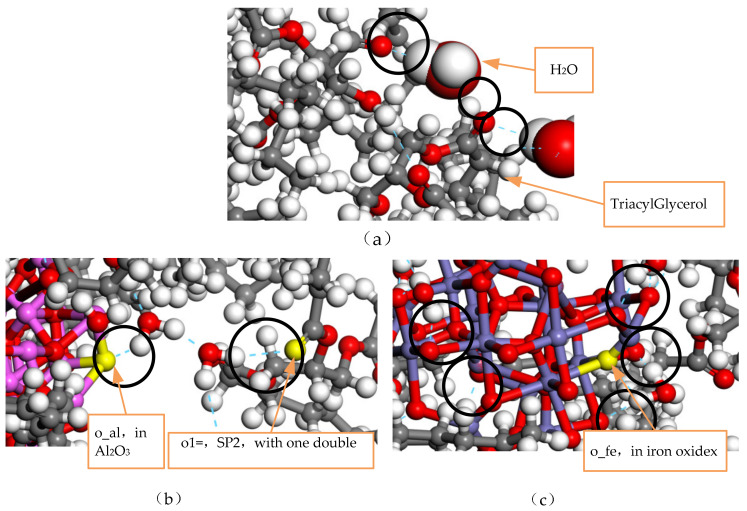
Type of hydrogen bonding in the model before and after the modification: (**a**) Model without added nanoparticles, (**b**) Al_2_O_3_ nano-modified model, (**c**) Fe_3_O_4_ nano-modified model.

**Figure 10 nanomaterials-13-00653-f010:**
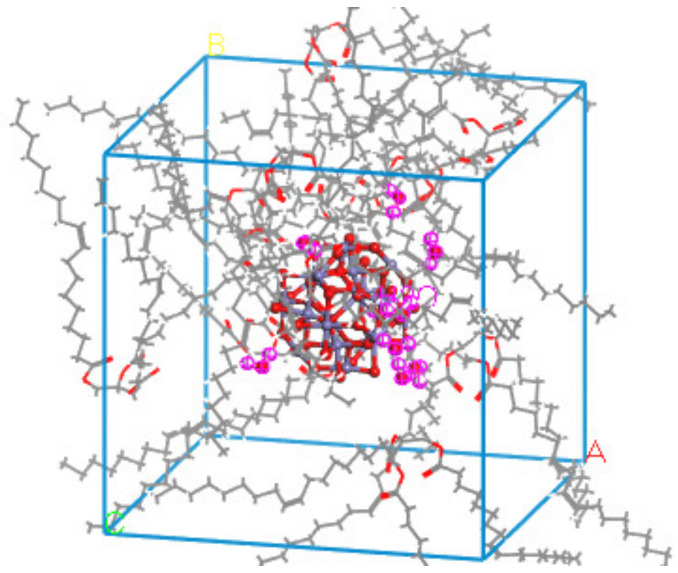
Distribution of water molecules in the nano-Fe_3_O_4_-modified model.

**Table 1 nanomaterials-13-00653-t001:** Fatty acid component content of common natural ester oil bases.

Name	Saturated Fatty Acids	Unsaturated Fatty Acids		
		Single	Double	Three
Rapeseed oil	7.9	55.9	22.1	11.1
Corn oil	12.7	24.2	58	0.7
Cottonseed oil	25.8	17.8	51.8	0.2
Olive oil	13.2	73.3	7.9	0.6
Soybean oil	14.2	22.5	51	6.8
Safflower oil	8.5	12.1	74.1	0.4
Sunflower oil	10.5	19.6	65.7	—

**Table 2 nanomaterials-13-00653-t002:** Interaction energy between metal oxide and insulating oil molecules at different temperatures (kcal/mol).

	NG-Fe_3_O_4_			NG-Al_2_O_3_		
	$E_{int}$	$E_{vdw}$	$E_{elec}$	$E_{int}$	$E_{vdw}$	$E_{elec}$
343 K	−1055.392961	−243.218	−777.387	−745.19325	−191.178	−517.464
363 K	−1007.773228	−225.794	−751.582	−746.300032	−188.846	−520.903
383 K	−1045.972345	−211.769	−799.415	−776.265149	−198.673	−541.04
403 K	−1069.692824	−224.039	−815.926	−773.84377	−191.542	−545.749
423 K	−1113.105939	−227.312	−865.217	−742.873264	−188.378	−517.945

**Table 3 nanomaterials-13-00653-t003:** Interaction energy between metal oxide and insulating oil molecules under the action of the electric field (kcal/mol).

	NG-Fe_3_O_4_			NG-Al_2_O_3_		
	$E_{int}$	$E_{vdw}$	$E_{elec}$	$E_{int}$	$E_{vdw}$	$E_{elec}$
343 K	−963.396727	−210.611	−718.029	−946.840935	−193.523	−716.767
363 K	−976.803808	−217.918	−724.098	−963.18838	−176.253	−750.384
383 K	−1017.271043	−237.76	−744.723	−961.921038	−173.297	−752.072
403 K	−1007.493808	−227.597	−750.238	−1022.619442	−186.397	−799.671
423 K	−935.869474	−211.89	−689.192	−927.643619	−170.895	−720.197

**Table 4 nanomaterials-13-00653-t004:** Diffusion coefficient of water molecules in the three groups of models.

	NG-1	NG-2	NG-3
343 K	0.025	0.002	0.001
363 K	0.046	0.008	0.006
383 K	0.272	0.025	0.011
403 K	0.065	0.061	0.034
423 K	0.166	0.121	0.105

**Table 5 nanomaterials-13-00653-t005:** Number and average bond length of hydrogen bonds in the three groups of models.

	NG-1	NG-2	NG-3
Number of hydrogen bonds	13	18	18
Average bond length of hydrogen bonds (Å)	1.96	1.82	1.68

## Data Availability

The study did not report any data.

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
