# Peer review of "Study on the Anti-Aging Performance of Different Nano-Modified Natural Ester Insulating Oils Based on Molecular Dynamics"

_nanomaterials, 2023, doi:10.3390/nano13040653_

Round 1

Reviewer 1 Report

"Study on the Anti-aging Performance of Different Nano-modified Natural Ester Insulating Oil Based on Molecular Dynamics" by Tao et al. studied the anti-aging performance of Fe3O4 and Al2O3 nano clusters modified natural ester insulating oil using molecular dynamics simulations. They found that the Fe3O4 modified composite is better by analyzing the water diffusions, free volume, hydrogen bonding, RDF, etc. However, there are flaws in methodology and inappropriate reasoning to support the results. More comments are as follows. 

1. The authors stressed several times in the introduction section on the "magnetic" Fe3O4 and "non-magnetic" Al2O3, yet nothings was done to investigate the magnetic properties or compare the results due to magnetic properties.

2. Except listing the previous findings that Fe3O4 has outstanding performance on the oil modification, the authors should also mention what the previous studies have done to explain the good performance. Had the previous studies done any analysis on the surface interaction, hydrogen bonding or RDF? What did they find and how can the authors compare the results with the previous studies?

3. Equation 2, the calculation of binding energy is inaccurate. It is not the negative value of the interaction energy. Binding energy should use the optimized structures when isolated, rather than using the conformation when the two components are interacting with each other.

4. Lots of unclear or inappropriate phrases. E.g.,

-- Abstract "Three models were established, two of which contained nano-Fe3O4 and nano-Al2O3 14 particle together with water concentration of 1 wt.%." 

-- Line 45, "chemical reflection"

-- Line 84, "analyze the anti-aging performance of two different properties of ..."

-- Line 99, " due to surface effects.". Did the authors actually study the surface effects?

-- Line 127-135, why mention canola oil?

-- Figure 2 is not the force field and charge setting.

-- What does it mean in Figure 3 the (4X4) superlattice surface model? 

-- Line 155, "periodically modified from 2D to 3D" doesn't make sense. How many layers of the surface? 

-- Line 186-187, I suggest to use potential energy rather than single point energy if the authors are averaging the energy across the simulation frames.

-- Line 219, what does 1% of water molecules mean?

-- Figure 6, too little information on the figure and hard to understand.

-- Line 231, what does "5 cycles" mean?

-- Line 238, does the "Nose" refer to "Nosé–Hoover"?

-- Line 270, "varies in the range of 0-50 Å^2" and the similar analysis followed, the authors should add the corresponding time range. Besides, such MSD range analysis is unimportant and all of the information can be extracted from the diffusion coefficient.

-- Line 275, how did the authors make the conclusion "indicating that the nat- 275 ural ester insulating oil was prone to aging at this temperature"? There is no inference provided.

-- Line 281-283, it's normal sense that water diffuses faster at higher temperature due to increased kinetic energy. 

-- Line 311, the authors didn't provide any support to prove the relation of diffusion and Ebind. Besides, the authors could delve into why NG-2 and NG-3 have slower diffusion.

-- Line 338, what does the "medium" mean in the sentence "The volume of the medium is relatively constant"

-- Line 379, "hydrogen bonding is a stronger non-bonding interaction ..." Stronger compared to what? 

-- Line 399, What is the definition of hydrogen bond (HB) in the manuscript? How many types of HB in the model? How many simulation frames have HB and how many total number of HB in each of those frame?  What is the averaged bond length and angle? How long does a HB last? How many H2O molecules in the model? The authors could also show a picture or two to interpret all types of the HB.

-- Line 414, wrong definition of the RDF regarding to the work in the manuscript.

-- Line 419, what is the purpose of the RDF analysis on all atoms?

-- Figure 13, there is no useful information extracted from this O-H RDF analysis.

-- Figure 14, cannot tell any difference from figure 14.

-- Line 481, the authors could study more on why there is more free volume in NG-2 and NG-3.

Author Response

Comments 1:The authors stressed several times in the introduction section on the "magnetic" Fe3O4 and "non-magnetic" Al2O3, yet nothings was done to investigate the magnetic properties or compare the results due to magnetic properties.

Response:The main purpose of the "magnetic" and "non-magnetic" references in the article is to highlight the differences between nanomaterials themselves. With the development of nanotechnology, there are more nano-materials that can be used to improve the performance of insulating oil, and we would classify them into magnetic nano-materials and non-magnetic nano-materials according to their nano-nature. The magnetic nanomaterials include Fe3O4, Fe2O3, TiO2, etc.; the non-magnetic nanomaterials include Al2O3, AIN, SiC and SiO2, etc. This paper selects different properties of nanomaterials to study the anti-aging performance of natural ester insulating oil, and explores the microscopic mechanism of nano-modification only as part of our researchs, and also as one of the bases for the selection of nanomaterials in the modification test stage of natural ester insulating oil. The magnetic nanomaterials will be further investigated for their specific properties and whether they will show a certain "targeting" in the actual operation.

Comments 2:Except listing the previous findings that Fe3O4 has outstanding performance on the oil modification, the authors should also mention what the previous studies have done to explain the good performance. Had the previous studies done any analysis on the surface interaction, hydrogen bonding or RDF? What did they find and how can the authors compare the results with the previous studies?

Response:Fe3O4 nanoparticles are added to natural ester insulating oil to form nano-insulating oil. The nano-particles in nano-insulating oil have a very high specific surface area and reaction activity, which can not only absorb the active oxygen molecules in the insulating oil to improve the oxidation resistance of the insulating oil, but also Fe3O4 magnetic nanoparticles have certain adsorption to the free metal particles in the insulating oil. Nanoparticles are solid dielectrics and insulating oil with nanoparticles has a higher thermal conductivity compared to crude oil; nanoparticles can absorb water in insulating oil, hinder the diffusion movement of water molecules and delay the ageing of insulating oil; nanoparticles will polarise under the action of electric field and form a large number of traps around them, thus hindering the migration of charged particles in the oil, reducing the conductive current of insulating oil and increasing the breakdown voltage; nanoparticles can also improve the matching coefficient between insulating oil and insulating paperboard. Nanoparticles can also improve the matching coefficient between the insulating oil and the insulating paper, thus improving the overall strength of the interface insulation of the oil-paper insulation system.

In a previous study, Fe3O4 nano-insulating oils were prepared and the electrical properties of the nano-insulating oils were tested experimentally.

Figure 1 Laboratory prepared nano-insulating oil

Table 1 Comparison of electrical properties of crude oil and nanoparticle modified insulating oils with different contents

Indicators

Blank sample

0.6% added nanoparticles

1% added nanoparticles

Dielectric loss factor

0.451%

0.453%

0.481%

Breakdown voltage

74kV

76kV

79kV

Dielectric constant

2.6

2.8

3.2

When analysing the ageing resistance of nano-insulating oils in a later study, surface interactions, hydrogen bonding etc. were analysed in order to analyse the effect of nanoparticles on water molecules in the nano-insulating oil model. This analysis is referenced to the research methods used to analyse the influence of the diffusion behaviour of water molecules in nano-Al2O3, SiO2, MgO and other modified mineral insulating oils.

Comments 3:Equation 2, the calculation of binding energy is inaccurate. It is not the negative value of the interaction energy. Binding energy should use the optimized structures when isolated, rather than using the conformation when the two components are interacting with each other.

Response:The equation for the interaction energy versus the binding energy in the text from the literature ”Wenyan Shi; Fengyun Wang; Mingzhu Xia.Molecular Dynamics Simulation of Interaction between Carboxylate Copolymers and calcite crystals.Acta Chimica Sinica 2006, 1817-1823.” This reference has been added to our paper.

Comments 4:Lots of unclear or inappropriate phrases. E.g.,

Comments 4.1:-- Abstract "Three models were established, two of which contained nano-Fe3O4 and nano-Al2O3 14 particle together with water concentration of 1 wt.%."

Response: Three models were established, an oil-water model without the addition of nanoparticles and two of which contained nano-Fe3O4 and nano-Al2O3 particles,where the concentration of water was 1wt.%. 

Comments 4.2:-- Line 45, "chemical reflection"  reaction

Response:The wrong word is used here, "chemical reflection" should be replaced by "chemical reaction"

Comments 4.3:-- Line 84, "analyze the anti-aging performance of two different properties of ..."

Response:The expression is unclear and the whole paragraph should be amended to read ”In this paper, molecular dynamics simulation technique is used to calculate the natural ester insulating oil before and after nano-modification, where the nano-materials are selected with reference to the literature [12] in which the non-magnetic material Al2O3 and the magnetic material Fe3O4 are chosen for comparative analysis to compare the performance of the two nano-modified oils and to explore the mechanism of nano-particle modification.”

Comments 4.4:-- Line 99, " due to surface effects.". Did the authors actually study the surface effects?

Response:The paper does not study surface effects, the expression here is inappropriate, the key word should be deleted, this paragraph is revised expression for “Metal nanomaterials have a large specific surface area, many surface active centres and increased surface active sites, which can cause nanoparticles and oil molecules to form a two-phase interface, affecting the overall performance of the nano-modified oil.”

Comments 4.5:-- Line 127-135, why mention canola oil?

Response:AS the paper refers to “The physicochemical and electrical properties of different vegetable oil-based natural ester insulating oils prepared were compared with those of mineral insulating oils, and it was found that the best overall performance of natural ester insulating oils prepared with higher content of monounsaturated fatty acids in vegetable oil base.” Therefore, rapeseed oil was chosen as the base oil for the preparation of natural ester insulating oils in this paper, which is explained here as a basis for modelling.

Comments 4.6:-- Figure 2 is not the force field and charge setting.

Response:the expression here is wrong and should be replaced by “Two forcefield types for Fe atoms in Fe3O4”.

Comments 4.7:-- What does it mean in Figure 3 the (4X4) superlattice surface model?

Response:The intention of showing Figure 3 in this paper is mainly to convey that the chemical bonding in the metal oxide has been removed before the layer structure is built up for calculation.

Comments 4.8:-- Line 155, "periodically modified from 2D to 3D" doesn't make sense. How many layers of the surface?

Response: "periodically modified from 2D to 3D" this phrase has been removed from the text. The surface thickness is 9.5Å. The entire paragraph has been amended as follows “The Build layers module in MS was used to construct the composite model of natural ester insulating oil and metal oxide.In layer1 the relaxed metal oxide (4X4) surface is selected, in layer2 the natural ester insulating oil amorphous monocell is selected, and in the Layer Detals module the vacuum layer of layer2 is increased to 30 Å. The two layer structure composite models are named NG-Fe3O4 and NG-Al2O3 respectively. The model structure is shown in Figure 4.”

Comments 4.9:-- Line 186-187, I suggest to use potential energy rather than single point energy if the authors are averaging the energy across the simulation frames.

Response: According to Literature “Tian, W.; Tang, C.; Wang, Q.; Zhang, S.; Yang, Y. The Effect and Associate Mechanism of Nano SiO2 Particles on the Diffusion Behavior of Water in Insulating Oil. Materials 2018, 11.” The original expression has been amended to read “ is the total energy of the interface model, is the potential energy values for metal oxides surface model, is the potential energy values for the insulating oil molecules model. ”

Comments 4.10:-- Line 219, what does 1% of water molecules mean?

Response: The text is unclear and has been amended in the original. The molar relationship is modelled by taking 100g of natural ester insulating oil above and converting its fatty acid ratio to a molar ratio. 1% water molecules means that the water content is 1% of the total natural ester insulating oil.

Comments 4.11:-- Figure 6, too little information on the figure and hard to understand.

Response: More information has been added to the picture and the statement about modelling in the text will be more detailed and revised as follows “In order to compare and analyse the ageing resistance of different nano-modified natural ester insulating oils, the natural ester insulating oils are to be aged before and after modification. Water molecules are one of the products of insulation ageing in transformers, and moisture has a important influence on the internal insulation property of transformers. We added the two nanoparticles mentioned in the previous section to the ageing insulating oil, used molecular dynamics simulation to investigate the effect of nanoparticles on water molecules in the insulating oil and to compare the modification effect of the two nanomaterials on the ageing insulating oil. Three models were built, oil-water model NG-1 without added nanoparticles, two sets of nano-modified models NG-2 and NG-3 with nano-Fe3O4 and nano-Al2O3 particles respectively added to the oil-water model, where the water content is 1% of the natural ester insulating oil content. The model construction process is shown in Figure 6, where the particle radius of the Al2O3 and Fe3O4 nanoparticle cluster model is 5Å.”

Figure 6. Construction process of the computational models before and after modification.

Comments 4.12:-- Line 231, what does "5 cycles" mean?

Response: The "5 cycle annealing treatment under NVE synthesis" is a simulated annealing cycle of 5 cycles of temperature increase and decrease.

Comments 4.13:-- Line 238, does the "Nose" refer to "Nosé–Hoover"?

Response: Yes, Themostat be selected as "Nose" when performing molecular dynamics calculations.

Comments 4.14:-- Line 270, "varies in the range of 0-50 Å^2" and the similar analysis followed, the authors should add the corresponding time range. Besides, such MSD range analysis is unimportant and all of the information can be extracted from the diffusion coefficient.

Response:The analysis of the MSD is rather cumbersome and poorly presented and the entire paragraph has been amended as follows “The simulation time is 200 ps for all three groups of models, and we only show the MSD curves for water molecules from 0 to 150 ps. It can be seen from figures that the range of motion of water molecules in the three groups of models is different at different temperatures. As the temperature gradually increases, there is a large difference in the range of motion of water molecules in the unmodified model and the two sets of nano-modified models. Figure 7(a) shows the MSD curves of the unmodified model NG-1. From the figure, it can be seen that when the temperature is 343 K, the mean square displacement size of water molecules all vary in the range of 0-50 Å2; when the temperature is 363 K~403 K, the mean square displacement of water molecules varies in the range of 0-200 Å2; when the temperature is 423 K, the mean square displacement of water molecules varies in the range of 0-300 Å2 varies. Figure 7 (b) and (c) shows the MSD curves of the Al2O3 and Fe3O4 nano-modified models respectively, compared with the unmodified model curves, the variation of the mean square displacement of water molecules in both nano-modified models is smaller than that of the NG-1 model at the same temperature, and the maximum range of motion does not exceed 100 Å2. 2 nanoparticle Al2O3-modified oil showed that the mean square displacement of water molecules varied in the range of 0-75 Å2; the mean square displacement of water molecules in the NG-3 nanoparticle Fe3O4-modified oil varied in the range of 0-50 Å2. At different temperatures from 343 K to 423 K, the mean square displacement of water molecules in model NG-3 varied less than that of model NG-2, indicating that the thermal stability performance of the Fe3O4 nanoparticle modified oil was better than that of the Al2O3 nanoparticle modified oil. This indicates that the diffusion of water molecules is relatively slow at low temperatures, and the diffusion range of water molecules gradually increases as the temperature increases. The addition of nanoparticles can effectively inhibit the effect of temperature on the diffusion of water molecules, thus reducing the risk of thermal ageing and enhancing the thermal stability of natural ester insulating oils.”

Comments 4.15:-- Line 275, how did the authors make the conclusion "indicating that the natural ester insulating oil was prone to aging at this temperature"? There is no inference provided.

Response: "indicating that the natural ester insulating oil was prone to aging at this temperature" have been deleted. The phrase is insufficient arguments in the original paragraph.

Comments 4.16:-- Line 281-283, it's normal sense that water diffuses faster at higher temperature due to increased kinetic energy.

Response:This phrase is lead to the conclusion later in the text “The addition of nanoparticles can effectively inhibit the effect of temperature on the diffusion of water molecules, thus reducing the risk of thermal ageing and enhancing the thermal stability of natural ester insulating oils.”

Comments 4.17:-- Line 311, the authors didn't provide any support to prove the relation of diffusion and Ebind. Besides, the authors could delve into why NG-2 and NG-3 have slower diffusion.

Response:The analysis of "Hydrogen Bonding" in the text is intended to explain the slower diffusion of water molecules in model NG-2 and NG-3.

Comments 4.18:-- Line 338, what does the "medium" mean in the sentence "The volume of the medium is relatively constant"

Response:The intention was to convey that the lattice volume of the model is relatively constant when performing molecular dynamics analysis, but the logic of direct modification is confusing, so the phrase "The volume of the medium is relatively constant" has been deleted outright.

Comments 4.19:-- Line 379, "hydrogen bonding is a stronger non-bonding interaction ..." Stronger compared to what?

ResponseCompared to van der Waals interaction and electrostatic interaction

Comments 4.20:-- Line 399, What is the definition of hydrogen bond (HB) in the manuscript? How many types of HB in the model? How many simulation frames have HB and how many total number of HB in each of those frame?  What is the averaged bond length and angle? How long does a HB last? How many H2O molecules in the model? The authors could also show a picture or two to interpret all types of the HB

.Response: Hydrogen Bond is defined as“Hydrogen Bond is the covalent bonding of an H atom to an X atom with a large electronegativity. If another atom Y with a large electronegativity and a small radius is present, a special intramolecular or intermolecular interaction is formed between X and Y using the H atom as a medium to produce X-H…Y. ”

NG-1

NG-2

NG-3

Number of hydrogen bonds

13

18

18

Average bond length of hydrogen bond(Å)

1.96

1.82

1.68

The main types of hydrogen bonding in the two modified models have been labelled in Figure 10 in the manuscript; hydrogen bonds are created between the water molecule and the water molecule, between the water molecule and the natural ester insulating oil, and between the water molecule and the H atom and the O atom in the nanoparticle. Hydrogen bonds are made in each frame of the simulation and the number of hydrogen bonds changes as the molecules move, Figure 10 shows the formation of hydrogen bonds in the last frame.

Table 2. Number and average bond length of hydrogen bonds in the three groups of models.

The models contain 10 water molecules, which can create at least 20 hydrogen bonds, but the number of hydrogen bonds does not reach 20 due to the constant movement of the water molecules. During playback of the simulated trajectories it was found that the water molecules in model NG-1 had the largest range of motion, while the two nanomodified models NG-2 and NG-3 had water molecules almost surrounding the nanoparticles.

Comments 4.21-4.24:-- Line 414, wrong definition of the RDF regarding to the work in the manuscript.

-- Line 419, what is the purpose of the RDF analysis on all atoms?

-- Figure 13, there is no useful information extracted from this O-H RDF analysis.

-- Figure 14, cannot tell any difference from figure 14.

.Response: The method used incorrectly, the section 3.3.4 RDF analysis work has been removed from the paper.

Comments 4.25:-- Line 481, the authors could study more on why there is more free volume in NG-2 and NG-3.

Response: the free volume fractions of the nano-models of NG-2 and NG-3 are smaller at the same temperature, and the growth rate of the free volume fraction with the increase of temperature is smaller than that of model NG-1, which indicates that the nanoparticles slow down the effect of temperature on the diffusive motion of water molecules in insulating oil.

Reviewer 2 Report

In this paper Authors investigated two different types of nanoparticles to modify insulating oils. The Authors studied the microscopic mechanistic of nano-modified models using molecular simulation techniques. Three models were established, two of which contained nano-Fe3O4 and nano-Al2O3 particle together with water concentration of 1 wt.%. The Authors stated that the diffusion of water molecules in the nano-modified model was slow, and the water molecules generated from transformer insulation aging were adsorbed around the nanoparticles, which inhibited the diffusion of water molecules, reduced the hydrolysis of ester molecules, and effectively enhanced the anti-aging performance of insulating oil. Below I presented some remarks that came to my mind during reading:

1.     Lines 32-35: These lines are missing the confirmation (citing articles) that mineral oils have low fire and flash points, etc. Please refer to the latest literature.

2.     Lines 46-50: Please present several literature items that say that the ability of natural esters to transport heat is worse than that of pure mineral oil. One should also quote the latest literature on the modification of natural esters in order to improve their thermal properties - e.g. viscosity, thermal conductivity, specific heat, etc.

3.     In my opinion the Introduction must be improved. In a research paper, it is expected that introduction section briefly explains the starting background and, even more important, the originality (novelty) and relevancy of the study is well established. Introduction should adequately represent the state of knowledge and clearly specify the purpose and motivation of taking up the topic. The area of research must be introduced with details for unfamiliar readers. The Authors should state what is special, unexpected, or different in their approach. I consider that the manuscript under review will benefit if the authors make all of these aspects as clear as possible to the readers. Moreover, the Authors do not clearly discuss their contribution in the introduction, before entering their main proposed research work as presented in the following chapters.

4.     The Authors should present the findings also highlighting current limitations of their study, and briefly mention some precise directions that they intend to follow in their future research work.

5.     Conclusions should be worded slightly different. Try to emphasize novelty. The Authors should highlight what are the advantages and disadvantages when comparing solutions from the scientific literature. Put some quantifications, and comment on the limitations. This is a very common way to write conclusions for a learned academic journal. The conclusions should highlight the novelty and advance in understanding presented in the work. In the Conclusions also, it would be useful to add information on further research of the authors related to the continuation of this research topic.

Author Response

Comments 1:Lines 32-35: These lines are missing the confirmation (citing articles) that mineral oils have low fire and flash points, etc. Please refer to the latest literature.

Response: The original text has been amended to read “However, the mineral insulating oil is difficult to degrade and environmentally unfriendly, and non-renewable, indicating the obvious unsustainability of using mineral insulating oil as the main transformer oil.”

Comments 2:Lines 46-50: Please present several literature items that say that the ability of natural esters to transport heat is worse than that of pure mineral oil. One should also quote the latest literature on the modification of natural esters in order to improve their thermal properties - e.g. viscosity, thermal conductivity, specific heat, etc.

Response: Relevant research references have been added to the original text

Comments 3:In my opinion the Introduction must be improved. In a research paper, it is expected that introduction section briefly explains the starting background and, even more important, the originality (novelty) and relevancy of the study is well established. Introduction should adequately represent the state of knowledge and clearly specify the purpose and motivation of taking up the topic. The area of research must be introduced with details for unfamiliar readers. The Authors should state what is special, unexpected, or different in their approach. I consider that the manuscript under review will benefit if the authors make all of these aspects as clear as possible to the readers. Moreover, the Authors do not clearly discuss their contribution in the introduction, before entering their main proposed research work as presented in the following chapters.

Response:The introductory section has been reworded to read “Transformer insulating oil is used in oil-immersed transformers [1] for arc extinguishing, cooling, heat dissipation and insulation, etc. ……However, the mineral insulating oil is difficult to degrade and environmentally unfriendly, and non-renewable, indicating the obvious unsustainability of using mineral insulating oil as the main transformer oil.……In order to investigate the effect of magnetic Fe3O4 nanoparticles on the performance of aged natural ester insulating oil, the insulating oil is first aged. Water molecules can degrade the insulation performance within the transformer, and adding a certain percentage of water molecules to the crude oil can accelerate the aging of the insulating oil. In this paper, molecular dynamics simulation technique is used to calculate the natural ester insulating oil before and after nano-modification, where the nano-materials are selected with reference to the literature [12] in which the non-magnetic material Al2O3 and the magnetic material Fe3O4 are chosen for comparative analysis to compare the performance of the two nano-modified oils and to explore the mechanism of nano-particle modification.”

Comments 4:The Authors should present the findings also highlighting current limitations of their study, and briefly mention some precise directions that they intend to follow in their future research work

.Response: This paper selects different properties of nanomaterials to study the anti-aging performance of natural ester insulating oil, and explores the microscopic mechanism of nano-modification only as part of our researchs, and also as one of the bases for the selection of nanomaterials in the modification test stage of natural ester insulating oil. The magnetic nanomaterials will be further investigated for their specific properties and whether they will show a certain "targeting" in the actual operation.

Comments 5:Conclusions should be worded slightly different. Try to emphasize novelty. The Authors should highlight what are the advantages and disadvantages when comparing solutions from the scientific literature. Put some quantifications, and comment on the limitations. This is a very common way to write conclusions for a learned academic journal. The conclusions should highlight the novelty and advance in understanding presented in the work. In the Conclusions also, it would be useful to add information on further research of the authors related to the continuation of this research topic.

Response: The analysis of the calculation results and the conclusions have been reformulated in more detail, for example,analysis of hydrogen bonding results “Hydrogen Bond is the covalent bonding of an H atom to an X atom with a large electronegativity. If another atom Y with a large electronegativity and a small radius is present, a special intramolecular or intermolecular interaction is formed between X and Y using the H atom as a medium to produce X-H…Y. Where X and Y represent non-metallic atoms with large electronegativity and small atomic radius, such as O atoms, F atoms, N atoms. As can be seen in Figure 10, hydrogen bonds are generated in the three models mainly between water molecules and water molecules, between water molecules and triacylglycerol, and between water molecules and nanoparticles, generating O-H…O. Figure 10(a) shows the types of hydrogen bonds in the unmodified oil-water model, with 13 hydrogen bonds generated in the last frame of the simulation, when nanoparticles are not added to the model, mainly generated by the H atom in the water molecule and the O atom in the triacylglycerol. The hydrogen bonds generated by water molecules and triacylglycerol form a special intermolecular interaction, which accelerates the hydrolysis reaction of ester molecules in natural ester insulating oil, thus reducing the insulating property of natural ester insulating oil.

During playback of the simulated track it was found that the water molecules in model NG-1 had the largest range of motion, while the water molecules in the two nanomodified models NG-2 and NG-3 almost surrounded the nanoparticles. Figure 10 (b) and (c) respectively shows the types of hydrogen bonds in the nano-Al2O3 and nano-Fe3O4 modification models, and it can be seen that hydrogen bonds are generated between the water molecules and the nanoparticles. Comparing the types of hydrogen bonding in the two modified models, model NG-2 shows that the H atoms in the water molecules form hydrogen bonds with the O atoms in the nanoparticles as well as with the O atoms in the natural ester insulating oil. On the other hand,the model NG-3 in the last frame type of hydrogen bonding is seen almost exclusively between the H atoms in the water molecules and the O atoms in the nanoparticles. Therefore, both the type of hydrogen bond and the bond length size of the hydrogen bond differ, with the average bond length of the hydrogen bond being shorter in the nano-Fe3O4 model. The obvious effect of the O atoms in the nanoparticles forming hydrogen bonds with the H atoms in the water molecules in the NG-3 model ,and indicates that the Fe3O4 nanoparticles have a greater influence on the water molecules and is the reason why the Fe3O4 nanoparticles are better able to inhibit the diffusion of water molecules. Figure 11 shows the distribution of water molecules in the nano-Fe3O4 modified model at the 101st ps of the simulation.” and the last paragraph of the conclusions has been revised to read “(3) By calculating the number of hydrogen bonds and the average bond length of the two modified models, it was found that the number of hydrogen bonds between water molecules and other molecules was the same in the two modified models, but the types of hydrogen bonds and the average bond length of hydrogen bonds were different. In the nano-Fe3O4 model, the O atoms in the nanoparticles form hydrogen bonds with the H atoms in the water molecules, and the bond lengths are shorter, indicating that the Fe3O4 nanoparticles have more influence on the water molecules”etc.

Round 2

Reviewer 1 Report

The revised version is much more clear. There are, however, a few issues that the authors should address including a major mistake on the binding energy concept. This manuscript should be reviewed again before being accepted.

-- Response 2. The authors could add some comments from Response#2, i.e., why Fe3O4 is good, along with the Fe3O4 references in the introduction section.

-- Response 3. It is incorrect for equation 2 to treat binding energy as the negative of interaction energy. The energies of the polymer and the surface should be calculated separately in their isolated states. Many studies have done this, e.g., 

-- "Son JH, Rybolt TR. Force field based MM2 molecule-surface binding energies for graphite and graphene. Graphene. 2013 Jan 24;2(01):18." See Equation 1 "where Em is the energy of an isolated gas phase molecule, Es is the energy of the isolated surface adsorbent material, and Ems is the energy of the molecule and solid surface system where the molecule is placed on the surface to represent the adsorbed state"

-- "Duflot D, Toubin C, Monnerville M. Theoretical Determination of Binding Energies of Small Molecules on Interstellar Ice Surfaces. Frontiers in Astronomy and Space Sciences. 2021:24.". See first equation in the METHOD section, "Where Eads is the total calculated energy of the species adsorbed on the surface while Esubstrate and Especies are obtained separately."

-- "Xu L, Lin J, Bai Y, Mavrikakis M. Atomic and molecular adsorption on Cu (111). Topics in Catalysis. 2018 Jun;61(9):736-50."See equation 1 "where Etotal is the total energy of the entire adsorbate-slab system, Esubstrate is the total energy of the clean Cu(111) slab by itself, and Egas-phase adsorbate is the total energy of the isolated adsorbate in the gas phase. "

-- "Zhu C, Yang G. Insights from the adsorption of halide ions on graphene materials. ChemPhysChem. 2016 Aug 18;17(16):2482-8." See equation 2

-- "Ghasemi AS, Mashhadban F, Ravari F. A DFT study of penicillamine adsorption over pure and Al-doped C60 fullerene. Adsorption. 2018 Jul;24(5):471-80." See equation 1

The binding energies should be re-calculated and the analysis on the results should be updated.

-- Response 4.10. If 1% is the weight percentage, it is better to use 1 wt%.

-- Response 4.11. The 2 nanomaterials on the very left in Figure 6 should also have annotations added.

-- Response 4.13. The correct terminology "Nosé–Hoover" should be used in the manuscript.

-- Response 4.20. The authors should add more information to the hydrogen bond. e.g., what is the O-O distance, O-H distance in the hydrogen bond definition? What is the O-O-H angle definition? Did Table 3 only show hydrogen bonds of one single frame? If not, could the authors add the uncertities (e.g., standart deviations) to the values?

Author Response

Dear Reviewer:

Thank you for comments concerning our manuscript entitled “Study on the Anti-aging Performance of Different Nano-modified Natural Ester Insulating Oil Based on Molecular Dynamics” (Manuscript ID: nanomaterials-2165823). Those valuable comments made the manuscript more complete, logical, and linguistically accurate. We have studied your comments carefully and have made corrections which we hope to meet with your approval. The revised portions are marked in blue in the manuscript. The main corrections in the paper and the responses to the reviewer’s comments are listed as follows:

Reviewer #1:

Comments 1:-- Response 2. The authors could add some comments from Response#2, i.e., why Fe3O4 is good, along with the Fe3O4 references in the introduction section.

Response:Most of the references given in the manuscript when introducing the study of nano-modified oil properties are related to Fe3O4 nano-modification, and in addition, the literature related to the study of the effect of magnetic field environment on Fe3O4 nano was added to the manuscript and modified as follows. “literature [12] investigated the structural evolution of cement paste with nano-Fe3O4 under magnetic field, and found the influence law of nano-fe3o4 particles on the structure formation of cement paste;the properties and relaxation times of Fe3O4, ZnO, Al2O3, and SiO2 nanoparticle materials were summarized in the literature [13]”

Comments 2:-- Response 3. It is incorrect for equation 2 to treat binding energy as the negative of interaction energy. The energies of the polymer and the surface should be calculated separately in their isolated states. Many studies have done this, e.g.,

Response:The thesis calculates the binding energy mainly to compare the stability of two composite models, but the data evidence given is insufficient, the logic is not rigorous enough, and the formula is wrong, this section is reworked as follows: “

  1. Interaction of Natural Ester Insulating Oil with Metal Oxide Surfaces

Metal nanomaterials have the characteristics of large specific surface area, many surface active centers and increased surface active sites. After doping the metal nanoparticles into the natural ester insulating oil, the nanoparticles and the insulating oil molecules will form a two-phase interface, and the interfacial interaction energy has a great influence on the stability of the nano-modified oil [19]. If the interaction energy between the surface of insulating oil molecules and the surface of nanoparticles is positive, it means that the two substances are incompatible; if the interaction energy is negative, it indicates that the two substances are attracted to each other and the interface is compatible. The larger the absolute value of the negative value of the interfacial interaction energy, the better the interfacial compatibility of the two substances and the better the stability of the nano-modified oil.

The modified nanomaterials Fe3O4 and Al2O3 selected in this paper are both metal nanomaterials. In order to initially determine whether the metal nanoparticles are compatible with the natural ester insulating oil, and to compare the stability of the two nano-modified oils. We established an interface composite model between metal oxide and natural ester insulating oil, and calculated the interaction energy between insulating oil molecules and metal oxide interface.

……

……

……

In order to reduce the calculation error caused by the different size of the model surface, the lengths of the model dimensions U and V need to be viewed after cutting the crystal surface, where the parameters U and V have equal values. Cutting the (1 0 0) crystal surface of unit cell Fe3O4 and the (0 0 -1) crystal surface of unit cell Al2O3, the thickness of the surface layer is 13Å, and viewing the lengths of model U and V is shown in Figure 3. Since The surface area of the original unit cell cut surface is small, the calculated surface area is increased according to the lengths of the surface model dimensions U and V. Supercell surface models of Fe3O4 (4×4) and Al2O3 (5×5) are respectively established for the construction of composite interface models.

Figure 3. Lengths of the original unit cell cut surface U and V: (a) Fe3O4 surface model, (b) Al2O3 surface model.

2.3. Construction of Layer Structure Model Building

The Build layers module in MS was used to construct the surface composite model of natural ester insulating oil and metal oxide.In layer1 the relaxed metal oxide (4×4) surface was selected, in layer2 the natural ester insulating oil amorphous monocell was selected, add the vacuum layer of layer2 to 30Å in the Layer Detals module, and name this composite model NG-Fe3O4. Repeat the above operation to construct another composite model named NG-Al2O3 by selecting the relaxed Al2O3 (5×5) surface in layer1.The model structure is shown in Figure 4.

Figure 4. Composite structure models: (a) Fe3O4 with insulating oil layer structure model NG-Fe3O4 , (b) Al2O3 with insulating oil layer structure model NG-Al2O3.

2.4. Simulation Details

In this paper, when performing molecular dynamics calculations on the constructed layer structure model, model optimization is to be performed first, using Fix Cartesian position function to fix all atoms of the metal oxide, using COMPASSII force field with Max.iteration set to 5000 when optimizing the structure. Keeping the metal oxide atoms still fixed, the optimized layer structure model was calculated by molecular dynamics, using NVT regular system synthesis with an integration step of 1fs.The transformer is mainly affected by temperature and electric field during operation [24]. In order to investigate the influence of temperature and electric field on the stability of the composite model, and to compare the stability differences between the two composite models of NG-Fe3O4 and NG-Al2O3, it is necessary to set reasonable temperature and electric fields for molecular dynamics calculations. The actual operation of the transformer, its internal oil temperature is approximately between 70°C and 120°C [25], and the hot spot temperature can be 120°C to 140°C. Therefore, the temperature range is set to 70°C to 150°C, which is converted to a thermodynamic temperature of 343 K to 423 K with an interval of 20 K. The molecular dynamics calculations are performed for the two sets of composite models at five temperatures. Since the simulation environment did not consider the possible breakdown of the material in actual operation, and to shorten the simulation time, the simulated electric field strength was set to an electrostatic field of 1010 V/m with the electric field direction along the positive direction of the Z axis [26]. the Forcite module could not perform the electric field calculation directly, and it needed to be run using a script edited in the perl language. The stability of the interfacial composite model of metal oxide and natural ester insulating oil molecules is related to the interaction energy between the two substances forming the interface, and the interfacial interaction energy is calculated as shown in Equation (1).

This is example 1 of an equation:

(1)

is the interaction energy between metal oxides and insulating oil molecules, is the total energy of the interface model, is the potential energy values for metal oxides surface model, is the potential energy values for the insulating oil molecules model [27].

2.5. Results and Discussion

We subjected the composite layer structure models NG-Fe3O4 and NG-Al2O3 to molecular dynamics calculations and analyzed the results. Table 2 gives the calculation results of the interaction energy between the two model molecules at different temperatures without applied electric field, and all the values of interaction energy are negative, indicating that the two metal oxides selected in this paper are compatible with the insulating oil molecules. In the table, is the total interaction energy of the two composite layer structure models,  is the van der Waals interaction energy, and  is the electrostatic interaction, from the results, 95% of the interaction energy between metal oxide and insulating oil molecules comes from intermolecular van der Waals force and electrostatic force, and the electrostatic interaction energy is dominant.

Table 2. Interaction energy between metal oxide and insulating oil molecules at different temperatures (kcal/mol) 

NG-Fe3O4

NG-Al2O3

343K

-1055.392961

-243.218

-777.387

-745.19325

-191.178

-517.464

363K

-1007.773228

-225.794

-751.582

-746.300032

-188.846

-520.903

383K

-1045.972345

-211.769

-799.415

-776.265149

-198.673

-541.04

403K

-1069.692824

-224.039

-815.926

-773.84377

-191.542

-545.749

423K

-1113.105939

-227.312

-865.217

-742.873264

-188.378

-517.945

Table 3. Interaction energy between metal oxide and insulating oil molecules under the action of electric field (kcal/mol)

NG-Fe3O4

NG-Al2O3

343K

-963.396727

-210.611

-718.029

-946.840935

-193.523

-716.767

363K

-976.803808

-217.918

-724.098

-963.18838

-176.253

-750.384

383K

-1017.271043

-237.76

-744.723

-961.921038

-173.297

-752.072

403K

-1007.493808

-227.597

-750.238

-1022.619442

-186.397

-799.671

423K

-935.869474

-211.89

-689.192

-927.643619

-170.895

-720.197

At different temperatures, the two model intermolecular interactions are affected by the temperature differently, the model NG-Fe3O4 interaction energy increases with the increase of temperature; the model NG-Al2O3 interaction energy increases first and then decreases with the increase of temperature. From Table 2, it is easy to find that the values of  in the two models do not change much with the change of temperature, indicating that the change of temperature has little effect on the compatibility stability of the composite model. At the same temperature, the values of the model NG-Fe3O4 interaction energy are much larger than those of the model NG-Al2O3.

The calculated results of the interaction energy between the two model molecules at different temperatures with the applied electric field strength of 1010 V/m are given in Table 3. Comparing the data in Table 2 and Table 3, it can be seen that both model intermolecular interaction energies changed after the application of electric field, with the electric field having a greater effect on the electrostatic interaction energy . The interaction energy between Fe3O4 and insulating oil molecules under the same temperature and electric field is still greater than that between Al2O3 and insulating oil molecules. From the change of data, the model NG-Al2O3 interaction energy increased by 30%, indicating that the composite model is more influenced by electric field; the model NG-Fe3O4 intermolecular interaction energy is reduced by less than 10% by electric field, indicating that the composite model is less influenced by electric field. In summary, the interaction between Fe3O4 and insulating oil molecules is stronger and the composite model is more stable.”

Comments 3:-- Response 4.10. If 1% is the weight percentage, it is better to use 1 wt%.

Response:The wrong word is used here, "1%" should be replaced by "1 wt%"

Comments 4:-- Response 4.11. The 2 nanomaterials on the very left in Figure 6 should also have annotations added.

Response:More information has been added to the picture

Comments 5:-- Response 4.13. The correct terminology "Nosé–Hoover" should be used in the manuscript.

Response: The wrong word is used here, "Nose" should be replaced by "Nosé–Hoover"

Comments 6:-- Response 4.20. The authors should add more information to the hydrogen bond. e.g., what is the O-O distance, O-H distance in the hydrogen bond definition? What is the O-O-H angle definition? Did Table 3 only show hydrogen bonds of one single frame? If not, could the authors add the uncertities (e.g., standart deviations) to the values?

Response: The RDF (RDF calculations were done in the original manuscript) of the O-H atomic pair is calculated after molecular dynamics calculations when the temperature is 363 K. The characteristic length of H...O is judged to be 0.22 nm; the angle of O-O-H differs according to the source of the two electronegative atoms, when the water molecule and the insulating oil molecule produce hydrogen bonds whose angle is 137°, and the hydrogen bond between the water molecule and the nanoparticle O-O- H angle is 96°, both of which are greater than 90°; Table 5 (In the original manuscript is Table 3) is the number and average bond length of hydrogen bonds in a single framework.

Sincerely,

nanomaterials-2165823

Reviewer 2 Report

All comments provided by me in the first review were included by the Authors in the revised version of the article. I think that in its current form there are no contraindications for publishing this article.

Author Response

Dear Reviewer:

Thank you for comments concerning our manuscript entitled “Study on the Anti-aging Performance of Different Nano-modified Natural Ester Insulating Oil Based on Molecular Dynamics” (Manuscript ID: nanomaterials-2165823). Those valuable comments made the manuscript more complete, logical, and linguistically accurate. 

Sincerely,

nanomaterials-2165823

Round 3

Reviewer 1 Report

The revised version is now more scientifically sound. There are only a few points/sentences require clarification or better grammar.

1. Line 108. "In order to initially determine ... two nano-modified oils. We established ... " Please change the period to comma and lowercase "We".

2. Line 175. "The Build layers module in MS was used..." Please elaboration what is "MS" .

3. Line 251. The authors first concluded "the model NG-Fe3O4 interaction energy increases with the increase of temperature; the model NG-Al2O3 interaction energy increases first and then decreases with the increase of temperature", then the authors had a different conclusion of "indicating that the change of temperature has little effect on the compatibility stability of the composite model." Please be consistent in the conclusion whether the temperature HAS or HAS NOT impact on the interaction energy.

Author Response

Dear Reviewer:

Thank you for comments concerning our manuscript entitled “Study on the Anti-aging Performance of Different Nano-modified Natural Ester Insulating Oil Based on Molecular Dynamics” (Manuscript ID: nanomaterials-2165823). The revised portions are marked in blue in the manuscript. The main corrections in the paper and the responses to the reviewer’s comments are listed as follows:

Comments 1: Line 108. "In order to initially determine ... two nano-modified oils. We established ... " Please change the period to comma and lowercase "We".

Response: “ .We ” has been revised to “,we” in the manuscript.

Comments 2:Line 175. "The Build layers module in MS was used..." Please elaboration what is "MS"

Response: The Build layers module in MS was used... has been revised to “The Build layers module in Materials Studio was used...” in the manuscript.

Comments 3: Line 251. The authors first concluded "the model NG-Fe3O4 interaction energy increases with the increase of temperature; the model NG-Al2O3 interaction energy increases first and then decreases with the increase of temperature", then the authors had a different conclusion of "indicating that the change of temperature has little effect on the compatibility stability of the composite model." Please be consistent in the conclusion whether the temperature HAS or HAS NOT impact on the interaction energy.

Response: Delete the paragraph “At different temperatures, the two model intermolecular interactions are affected by the temperature differently, the model NG-Fe3O4 interaction energy increases with the increase of temperature; the model NG-Al2O3 interaction energy increases first and then decreases with the increase of temperature.” in the original manuscript and amend the conclusion section to “ From Table 2, it is easy to find that the values of in the two models do not change much with the change of temperature, indicating that the change of temperature has little effect on the compatibility stability of the composite model. At the same temperature, the values of the model NG-Fe3O4 interaction energy are much larger than those of the model NG-Al2O3. The calculated results of the interaction energy between the two model molecules at different temperatures with the applied electric field strength of 1010 V/m are given in Table 3. Comparing the data in Table 2 and Table 3, it can be seen that both model intermolecular interaction energies changed after the application of electric field, with the electric field having a greater effect on the electrostatic interaction energy. The interaction energy between Fe3O4 and insulating oil molecules under the same temperature and electric field is still greater than that between Al2O3 and insulating oil molecules. From the change of data, the model NG-Al2O3 interaction energy increased by 30%, indicating that the composite model is more influenced by electric field; the model NG-Fe3O4 intermolecular interaction energy is reduced by less than 10% by electric field, indicating that the composite model is less influenced by electric field. In summary, the interaction between Fe3O4 and insulating oil molecules is stronger and the composite model is more stable.”

Sincerely,

nanomaterials-2165823
